# Culture Degeneration Reduces Sex-Related Gene Expression, Alters Metabolite Production and Reduces Insect Pathogenic Response in *Cordyceps militaris*

**DOI:** 10.3390/microorganisms9081559

**Published:** 2021-07-22

**Authors:** Peter A. D. Wellham, Abdul Hafeez, Andrej Gregori, Matthias Brock, Dong-Hyun Kim, David Chandler, Cornelia H. de Moor

**Affiliations:** 1Gene Regulation and RNA Biology Laboratory, Division of Molecular Therapeutics and Formulation, School of Pharmacy, University Park Campus, University of Nottingham, Nottingham NG7 2RD, UK; peter.wellham@nottingham.ac.uk (P.A.D.W.); Abdul.hafeez@nottingham.ac.uk (A.H.); 2Centre for Analytical Bioscience, Advanced Materials and Healthcare Technologies Division, School of Pharmacy, University Park Campus, University of Nottingham, Nottingham NG7 2RD, UK; Dong-hyun.kim@nottingham.ac.uk; 3Fungal Genetics and Biology Group, School of Life Sciences, University Park Campus, University of Nottingham, Nottingham NG7 2RD, UK; Matthias.brock@nottingham.ac.uk; 4Mycomedica d.o.o., Podkoren 72, 4280 Kranjska Gora, Slovenia; andrej.gregori@zanaravo.com; 5Warwick Crop Centre, School of Life Sciences, University of Warwick, Warwick CV35 9EF, UK; Dave.Chandler@warwick.ac.uk

**Keywords:** entomopathogen, degeneration, *Cordyceps militaris*, *Galleria mellonella*, metabolomics, fungal sexual biology

## Abstract

*Cordyceps**militaris* is an entomopathogenic ascomycete, known primarily for infecting lepidopteran larval (caterpillars) and pupal hosts. Cordycepin, a secondary metabolite produced by this fungus has anti-inflammatory properties and other pharmacological activities. However, little is known about the biological role of this adenosine derivate and its stabilising compound pentostatin in the context of insect infection the life cycle of *C. militaris*. During repeated subcultivation under laboratory conditions a degeneration of *C. militaris* marked by decreasing levels of cordycepin production can occur. Here, using degenerated and parental control strains of an isolate of *C. militaris*, we found that lower cordycepin production coincides with the decline in the production of various other metabolites as well as the reduced expression of genes related to sexual development. Additionally, infection of *Galleria mellonella* (greater wax moth) caterpillars indicated that cordycepin inhibits the immune response in host haemocytes. Accordingly, the pathogenic response to the degenerated strain was reduced. These data indicate that there are simultaneous changes in sexual reproduction, secondary metabolite production, insect immunity and infection by *C. militaris*. This study may have implications for biological control of insect crop pests by fungi.

## 1. Introduction

*Cordyceps militaris* (L.) (Hypocreales: Cordycipitaceae) is an obligately-killing, entomopathogenic fungus (EPF) that infects soil-dwelling insects, predominantly lepidopteran larval and pupal hosts [1]. It has a worldwide distribution, but is considered to be restricted to forest habitats. The fungus is well known for the production of cordycepin, (3′-deoxyadenosine) [2], an immune-modulating metabolite that has been the subject of pharmacological research particularly in recent years (e.g., [3,4]). Cordycepin is a polyadenylation inhibitor with anti-inflammatory properties [3,5]. Cordycepin biosynthesis is coupled with the production of pentostatin, which prevents cordycepin degradation by adenosine deaminase [6]. Genes essential for the biosynthesis are found at adjacent loci in the cordycepin biosynthesis gene cluster [6].

The natural function of cordycepin in the life cycle of *C. militaris* has not been widely studied. Because *C. militaris* must kill its host in order to complete its life cycle, there is likely to be a strong selection pressure for traits that enable high virulence [7]. Cordycepin has been proposed to act as the direct cause of host insect death after initial colonisation [8]. However, given its anti-inflammatory activity—and the fact that immune evasion is a key factor in successful infection by many pathogens [9]—a role for this metabolite in repression of the host immune system appeared likely to us [10]. Indeed, it has recently been demonstrated that the expression of immune genes in *Galleria mellonella* and *Drosophila melanogaster S2r+* cells was reduced by the addition of cordycepin. Additionally, when cordycepin was introduced to caterpillars, survival from infections with *C. militaris* and the related pathogen *Beauveria bassiana* was reduced [11]. Cordycepin has also been shown in another study not only to aid killing of the insect host by *C. militaris*, but also by other, non-cordycepin-producing fungal pathogens when injected with conidia [12].

In *C. militaris*, sexual development occurs after colonisation of an insect host, resulting in the emergence of a stroma (fruiting body) bearing asci. An asexual stage in which conidia are produced has been described [13], but its importance in the natural setting is unknown.

The greater wax moth, *Galleria mellonella* (Lepidoptera: Pyralidae), has been used as an infection model for a wide range of fungal pathogens (e.g., [14,15,16,17]). *G. mellonella* caterpillars are easy to maintain under laboratory conditions and can be used for injection by hand owing to their large size (reaching approximately 2 cm in length) [18]. The popularity of this model has led to a refinement of the rearing of the caterpillars and moths and standardised methods for their experimental use (e.g., [19]). Furthermore, the genome of this organism has been sequenced [20] and expression of immune-related loci in *G. mellonella* has been investigated in several studies [21,22,23], including in response to an entomopathogen [11,23].

The phenomenon of degeneration, in which cultures of *C. militaris* show a decline in the production of cordycepin, has been reported in multiple studies (e.g., [24,25]). This has been found to be simultaneously occurring with declines in the expression of genes involved in cordycepin synthesis, mating-type genes and chromatin structure genes [25]; as well as carotenoid content and amylase activity [24]. We hypothesise in this study that this degeneration represents a shift away from the characteristics of the sexual phase (teleomorph) towards the vegetative state and that this change coincides with the reduced ability of the fungus to infect insects. If the case, this could be an effect related to attenuation of related species *Beauveria bassiana* and *Metarhizium anisopliae*. In these species, successive subcultivation has been shown to result in reduced virulence to model hosts, which would cause decline of product viability in the case of bioinsecticides, for which these species are used [26,27].

To test our hypotheses, we conducted a series of experiments on an original, high cordycepin-producing isolate of *Cordyceps militaris*, *CM2*, from China—from which two strains were derived—the original parental control strain, and a degenerated strain, which arose from successive subcultivation on potato dextrose agar over a period of two years, after 14 passages. The *CM2* isolate is heterothallic, and the strains used in this study were monokaryotic, possessing the mating type locus *MAT 1-2-1*, providing an effectively clonal model. Cordycepin and pentostatin production, as well as general metabolite output, were compared in the two strains using targeted and untargeted liquid chromatography mass spectrometry (LC-MS)-based metabolite profiling. One week-old PDA (potato dextrose agar)-grown cultures were assessed. Gene expression analysis targeting genes involved in cordycepin synthesis and sexual development were also performed using reverse transcription and quantitative PCR (RT-qPCR).

To test the effects of degeneration on host pathogenic response to *C. militaris*, we tested the two strains in a *Galleria mellonella* infection model. Larvae (caterpillars) were treated with injections of fungal material with or without the addition of extracts from cultures of both strains, as well as pure cordycepin and pentostatin. In response to the injection treatments, assays of daily proportional responsiveness, melanisation and fungal emergence from caterpillars were performed. Melanisation—an insect immune response to infection—and fungal emergence—an observation of fungal proliferation in the host—were used to assess and compare host pathogenic response to the parental control and degenerated strains. Responsiveness observations gave indications of moribund status of caterpillars, but however were not indicative of the point of death of the host. To investigate the possible effects on cordycepin on larval hosts, the expression of immune-related genes in injected caterpillars were tested. The polysaccharide curdlan was used in these experiments as a mock fungal infection, stimulating immune responses, upon which the effect of added cordycepin was tested.

Our findings provide evidence that cordycepin inhibits the immune responses of host haemocytes; and that the degenerated strain of *C. militaris*—which has reduced cordycepin and pentostatin production—displays a reduced pathogenic response in the host.

## 2. Material and Methods

### 2.1. Fungal Culture Media

The parental control *Cordyceps militaris CM2* isolate was originally cultured from stroma found in China. The *CM2* isolate was successively subcultivated on potato dextrose agar over a period of two years, through 14 passages, which resulted in a degenerated strain. Before experiments, both the parental control strain and the degenerated strain were revived from stock cultures stored at −80 °C. Using ITS primers (ITS1: CTTGGTCATTTAGAGGAAGTAA and ITS4: TCCTCCGCTTATTGATATGC) [28], the species identity of *C. militaris* was confirmed. RAPD PCR (S62 primer: GTGAGGCGTC [24]) confirmed the parental control and degenerated strains to originate from the same isolate (See Appendix A). Experimental plates and liquid cultures were prepared by transfer from initial revival potato dextrose agar (PDA) plates. Potato dextrose broth (PDB) was used for preparation of liquid and solid plate cultures respectively. All solid cultures were grown on experimental plates for two weeks before sample extraction. All cultures were grown at 25 °C, in the dark, with liquid cultures grown under static conditions.

### 2.2. Reverse Transcription and Quantitative PCR (RT-qPCR)

RNA extractions from fungal material from experimental solid culture plates were prepared. This was conducted using the GeneJET^®^ RNA Purification Kit (Thermo Fisher Scientific, Hemel Hempstead, UK), according to manufacturers’ instructions—specifically a tailored version of the protocol for yeast extractions. Reverse transcription was performed using SuperScript^®^ III Reverse Transcriptase (Thermo Fisher Scientific, Hemel Hempstead, UK), according to manufacturers’ instructions, using 200 ng total RNA in each sample. Quantitative PCR was carried out using GoTaq^®^ 2X qPCR Mastermix (Promega UK, Southampton, UK), according to manufacturers’ instructions. A Qiagen^®^ rotor qPCR machine was used, with SYBR^®^ green. A PCR programme of 10 min at 90 °C, then forty repeated cycles of 30 s at 95 °C, 30 s at 58 °C and 30 s at 72 °C was used.

*C. militaris* primers for this purpose were designed to amplify in each case a region of approximately 200 base pairs from cDNA, and *G. mellonella* qPCR primers were used from previous studies [21,22,23]. All primer sequences are listed in Table 1. Primers were validated using serial DNA dilutions and specificity of amplification was verified by separating products on an agarose gel.

### 2.3. Liquid Chromatography-Mass Spectrometry (LC-MS)

An LC-MS system—Accela LC and Exactive Orbitrap MS (Thermo Fisher Scientific, Hemel Hempstead, UK)—was used for metabolomic analysis of degenerated and parental control strains of *CM2*.

#### 2.3.1. Fungal Sample Extraction

For solid cultures, each sample was prepared from 30 mg of fresh fungal material removed from a PDA plate, after one week of growth. Seven biological replicates for each treatment group were used in LC-MS analysis. A freeze-thaw method using liquid nitrogen was used for the preparation of fungal samples, with a solvent ratio of 1:1:2 of methanol:water:chloroform [29]. Following 24-h storage at −20 °C, the samples were centrifuged and the supernatant was taken for further analysis.

#### 2.3.2. Liquid Chromatography

The ZIC^®^-*p*HILIC (Merck Sequant, Watford, UK) column (4.6 mm × 150 mm, particle size 5 µm) was used for chromatographic separation which was performed on an Accela system (Thermo Fisher Scientific, Hemel Hempstead, UK). Mobile phases were 20 mM ammonium carbonate in water (phase A, pH 9.1) and acetonitrile (phase B). A 24 min-per-sample gradient system was used, with a flow rate of 300 µL/min, at a column temperature of 45 °C. With a starting gradient of 20% A, increasing to 95% A over 15 min, an equilibration followed to give the 24 min run time per sample. Samples were maintained at 4 °C and an injection volume of 10 µL was used.

#### 2.3.3. Mass Spectrometry

The Accela LC system was coupled to an Exactive (Thermo Fisher Scientific, Hemel Hempstead, UK) Orbitrap mass spectrometer, with an electrospray ionisation (ESI) ion source. Both positive and negative ESI full-scan modes were used as conducted previously [29]. An automatic gain control of 3 × 10^6^ was used and the resolution was 70,000 with an m/z range 70 to 1050. Prior to use, each time the system was calibrated with positive and negative calibration mixes. In all runs, quality control (QC) samples generated by mixing an equal volume of each sample and both spiked and non-spiked with standards were injected to assess the instrument performance. In targeted runs, standards of cordycepin and pentostatin, at set concentrations 1, 5, 10, 25, 50 and 100 µM were used. The sequence of samples injected followed the previous recommendations [30]. Five mixtures (named A, B, C, D and E) containing authentic standards of 250 metabolites were used for the untargeted run, as described previously [29]. Untargeted data was analysed using XCMS [31], MzMatch [32], IDEOM [33] and SIMCA-P (Sartorius, Göttingen, Germany); and targeted data was analysed using Tracefinder (Thermo Fisher Scientific, Hemel Hempstead, UK) software, to quantify levels of cordycepin and pentostatin in the samples. Accurate mass and retention time based on authentic standards were used for identification of metabolites (Metabolomics Standards Initiative level 1 identification). When authentic standards were not used, putative identification was performed using accurate mass and predicted retention times (Metabolomics Standards Initiative level 2 identification) [34,35].

A similar method was used for LC-tandem MS (LC-MS/MS) on a Q-Exactive Plus MS equipped with Dionex U3000 UHPLC system (Thermo Fisher Scientific, Hemel Hempstead, UK), to confirm the identities of cordycepin and pentostatin by fragmentation produced using MS/MS (Metabolomics Standards Initiative level 1 identification) (see Appendix A).

### 2.4. Galleria mellonella Caterpillar Treatments

In all cases, *G. mellonella* larvae (caterpillars) were reared until final instar stage and size of 1.8–2.2 cm (approximately 250–290 mg) before selection for experimental treatment. Injections of 100 µL were given to each caterpillar behind the last left proleg, using sterile 0.3 mL 30 g 8 mm BD Microfine syringes and needles (UKMEDI). The reason for using 100 µL injections rather than a lower volume was the accuracy with which this larger volume could be injected by hand into large numbers of caterpillars.

In the first of these experiments, the injections consisted of conidial suspensions (100,000 spores per caterpillar) and spent liquid media, both from four-week old *C. militaris CM2* strain potato-dextrose broth (PDB) cultures. These cultures were grown under static conditions, in the dark, at 25°C. The high dose of spores was selected to expose the potential contrast between effects of the parental control and degenerated strains.

In the second experiment, the injections consisted of phosphate buffered saline (PBS)-washed spore suspensions (1000 spores per caterpillar) in PBS, both with or without additions of 50 µM cordycepin and 1.5 nM pentostatin both separately, and in combination. The lower spore dose compared to the first injection experiment was chosen to enable the detection of possible effects from added cordycepin and pentostatin. In both experiments, the parental control and degenerated strains were used, as well as three control treatments, consisting of unused sterile PDB media only or PBS only (blank), an uninjected control, and a needle control. Cordycepin and pentostatin-only (in PBS) controls were also used in the second experiment. Injection suspensions were prepared in triplicate to limit possible confounding effects from technical error.

Once injected, caterpillars were kept in petri dishes, in the dark and at a constant temperature of 22 °C, under ambient conditions and data recordings were taken daily. Caterpillars were reared on a diet of honey and bran mixture, but not fed after injection treatments. Observations were recorded by visual appearance and blinded to the treatment, through the assignment of random number codes for different dishes by a colleague. 

Responsiveness of caterpillars, melanisation, and emergence of fungal hyphae from caterpillars were observed by eye. Responsiveness was recorded by the observation of any physical self-movement when caterpillars were individually disturbed using tweezers. Therefore, caterpillars immobilised at the time of recording for any reason were marked as non-responsive. All three observations were recorded as a proportion of the total caterpillars included in the experiments from the start. Melanisation and fungal emergence, as well as responsiveness data were recorded from one day after injection onwards. These observations were chosen because they could be recorded with large numbers of caterpillars on a daily basis. Given the difficulty of determining the point of caterpillar death in these experiments, the observation of responsiveness was taken as an indication not of death but of moribund status of caterpillars. It should therefore be noted that unresponsive caterpillars were not necessarily dead. 

Occurring immediately upon detection of a fungal pathogen [36], melanisation of fungal tissues and surrounding areas is initiated by haemocytes, and is part of the wider insect immune response taking place in the haemocoel, along with encapsulation of invading fungal cells [37]. In melanisation, melanin is produced from the oxidation of phenolic molecules following the prophenoloxidase cascade [38]. This oxidisation is catalysed by active phenoloxidase, formed from prophenoloxidases via a serine protease cascade [39]. Phenoloxidase activity is significantly elevated within 12 hours of injection of fungal spores [40], and hence can form a rapid and observable immune response. Hence, melanisation was chosen as an observation for insect immune response to the fungus. However, melanin production can occur around wounds as well as in response to invading pathogens. As such, it was decided that only caterpillars showing melanisation of the entire body (rather than just around the injection site) would be recorded as melanised (see Figure 1). 

Each treatment group consisted of 60 caterpillars. Proportions of caterpillars showing full melanisation, fungal emergence or responsiveness in each case over time were plotted, with 95% confidence intervals. To compare between treatments, log-rank (Mantel-Cox) tests with Bonferroni corrections were performed. Prism software (GraphPad, San Diego, CA, USA) was used for statistical analyses and creating figures.

In gene expression experiments, caterpillars were frozen in liquid nitrogen and then briefly thawed before the removal of and RNA extraction from the haemolymph, two hours after injection. RNA extractions from cells in the removed haemolymph of *Galleria mellonella* were carried out using the ReliaPrep^®^ RNA Cell Miniprep System (Promega UK, Southampton, UK), according to manufacturers’ instructions. Reverse transcription was performed using SuperScript^®^ III Reverse Transcriptase (Thermo Fisher Scientific, Hemel Hempstead, UK), according to manufacturers’ instructions, using 200 ng total RNA in each sample. Quantitative PCR was carried out using GoTaq^®^ 2X qPCR Mastermix (Promega UK, Southampton, UK), according to manufacturers’ instructions. A Qiagen^®^ rotor qPCR machine was used, with SYBR^®^ green. A PCR programme of 10 min at 90 °C, then forty repeated cycles of 30 s at 95 °C, 30 s at 58 °C and 30 s at 72 °C was used.

### 2.5. Statistical Analysis

Statistical analyses of RT-qPCR analyses and targeted LC-MS were performed using GraphPad PRISM software—using t-tests to compare between means. When comparing between multiple groups, Bonferroni corrections were applied. Normality of these data sets was assessed using the Shapiro–Wilk normality test. Data from *Galleria mellonella* assays were analysed using the same software, with Kaplan Meier plots made for responsiveness data, and treatment groups compared using the log-rank (Mantel–Cox) test, with Bonferroni corrections. Untargeted data was analysed statistically using an orthogonal projections to latent structures discriminant analysis (OPLS-DA) model on SIMCA-P software, to obtain variable importance in the projection (VIP) values. Data from metabolites with VIP values > 1 were further analysed using t-tests with false discovery rate (FDR) corrections to find significant differences between the control and degenerated strain samples. A heatmap of significantly changed metabolites was constructed using R software (version 4.0.0), using the pheatmap package.

## 3. Results

### 3.1. Comparing Parental Control and Degenerated CM2 Strains

#### 3.1.1. Growth Rates of the Parental Control and Degenerated *C. militaris* Strains

To compare the characteristics of the two strains of *C. militaris*, we had to be sure that these strains are indeed genetically identical and that their vigour is similar. RAPD PCR screens confirmed the identity of both strains as from the same *C. militaris* isolate (see Appendix A). Growth rates of the parental control and degenerated strains were compared on PDA plates, and were found to not be significantly different (see Appendix A). Thus, the strains appear very similar in growth rate and DNA.

#### 3.1.2. Comparison of Cordycepin and Pentostatin Levels in Parental Control and Degenerated *C. militaris* Strains

We hypothesised that repeated subcultivation of *C. militaris* results in degeneration of the strain, marked by reduced production of cordycepin and its protector metabolite pentostatin. To test this, cordycepin and pentostatin levels were compared in the parental control and degenerated strains using LC-MS. The concentrations of cordycepin and pentostatin were significantly lower in the degenerated strain, as shown in Figure 2 (df = 4, t = 30.8, *p* < 0.0001). These data indicate that changes in secondary metabolite production have occurred during the repeated subcultivation of the culture.

#### 3.1.3. Differences in Metabolites Detected

Totals of 815 and 718 metabolites in control and degenerated samples, respectively, were either putatively or confidently identified. In addition to the lower levels of the secondary metabolites cordycepin and pentostatin, general LC-MS untargeted analyses detected significant changes in 60 metabolites in samples of the degenerated strain compared to those of the parental control strain, fulfilling a range of functions (Figure 3 and Figure 4). Of these significantly different metabolites, some involved in the citrate cycle and purine metabolism were affected (Figure 5), such as citrate, pyruvate and adenine—all of which were at a reduced level in the degenerated strain compared to the control strain. These findings imply a reduced energetic output of the degenerated strain (citrate cycle). Additionally, they are also consistent with the reduced output of cordycepin and pentostatin—given the proposed formation of cordycepin as part of the purine metabolism pathway via adenosine [6].

#### 3.1.4. Gene Expression Differences

As the production of cordycepin and pentostatin were reduced in the degenerated strains, we investigated whether these changes are reflected in gene expression. Furthermore, we investigated the expression of genes related to sexual reproduction, having hypothesised that sexual developmental might be an important driver for cordycepin production. To assess possible differences in the expression levels of genes relating to cordycepin production, sexual development and meiosis, assays of gene expression in *C. militaris* were carried out using reverse transcription PCRs (RT-qPCRs). Data was normalised to the combined mean values of control genes, *GAPDH*, *Actin*, *Calmodulin* and *rps3*. Figure 6 shows relative normalised mRNA levels of selected genes of interest in the parental control and degenerated strains. These genes of interest are the genes involved in the biosynthesis and export of secondary metabolites cordycepin and pentostatin (*Cns1-4*); and velvet protein-encoding and related genes involved in sexual development and the inhibition of asexual development. For the majority of the genes of interest considered, significantly lower levels of mRNA relative to the control genes were detected in degenerated compared to parental control strain samples. This was according to two-tailed t-tests (df = 4). This indicates lower secondary metabolite production and reduced sexual development and less potential for meiotic activity. In the case of the cordycepin/pentostatin biosynthesis cluster genes, this is consistent with the metabolomic findings of reduced production of cordycepin and pentostatin and the pathway of purine metabolism.

### 3.2. Effects on Galleria mellonella Caterpillars

#### 3.2.1. Immune-Related Gene Expression in Response to Cordycepin in Caterpillars

The differences between the parental control and degenerated strains of *CM2* in cordycepin production coincided with reduced expression of sex-related genes. Given that the sexual phase of this species manifests after colonisation of an insect host, we decided to test the effects of cordycepin on an insect host model. First, the effect of cordycepin was tested on *Galleria mellonella* caterpillars. Caterpillars were injected with curdlan, a mimic of fungal infection [41], to investigate the role of cordycepin in suppressing the insect immune response. Two hours after curdlan injection, the expression of the immune response-related genes *Lysozyme*, *Gallerimycin* and *Galiomicin* were tested in caterpillar haemocytes relative to the control gene *S7e*. Out of these, *Lysozyme* was found to be significantly upregulated with stimulation by curdlan compared to the control (uninjected) and control (blank) treatments, as shown in Figure 7 (df = 10, t = 9.124, *p* < 0.0001; df = 10, t = 4.803, *p* = 0.0072). There were however, no significant differences in expression for the other three genes (see Appendix A). When curdlan was injected together with 50 µM cordycepin, the upregulation of *Lysozyme* by caterpillar haemocytes was significantly reduced (Figure 7) (df = 10, t = 3.687, *p* = 0.042). This indicates a biological role of cordycepin in the inhibition of the immune activity of host haemocytes. These data indicate that the production of cordycepin by the fungus affects the insect immune response.

#### 3.2.2. Effects of Fungal Injections on Caterpillars

To further test the hypothesis that cordycepin plays a role in the host infection process, we decided to test if degeneration affects the pathogenic response of the host. *Galleria mellonella* caterpillars were injected with suspensions of conidia (100,000 spores) in liquid media from four-week old *C. militaris CM2* strain potato-dextrose broth (PDB) cultures, of both parental control and degenerated strains. The reason for the use of the spent media as well as spores was to include secondary and other metabolites released by the fungal cultures into the media.

Responsiveness (Figure 8) in control and degenerated strain-injected caterpillars was not significantly different, but both of these were significantly different to all controls (df = 1, X^2^ = 123.6, 117.5, 125.7, 130.6, 122.1, 132.8, *p* values < 0.001). All caterpillars which survived to the last time point ultimately pupated. Both fungal treatments showed significantly higher levels of full melanisation in caterpillars than all of the control treatments (df = 1, X^2^ = 127.8, 79.36, 127.8, 58.9, 22.9, 58.9, *p* values < 0.001). There were no cases of full melanisation in the control (uninjected) caterpillars or control (needle)-treated caterpillars at all. As expected, none of the control treatment caterpillars showed any fungal emergence. However, the parental control strain caused a significantly higher proportion of both fully melanised caterpillars (df = 1, X^2^ = 16.14, *p* = 0.001) and fungal emergence (df = 1, X^2^ = 14.6, *p* = 0.001) than the degenerated strain (Figure 9 and Figure 10). In summary, these results indicate that the degenerated strain displays reduced pathogenic responses in the host.

In a subsequent injection experiment, the effects of cordycepin and pentostatin in isolation and together on the infection of caterpillars from spores of the *CM2* strains grown in liquid cultures were tested. In each case, 1000 spores were injected into each caterpillar, previously washed in phosphate buffered saline (PBS) to remove fungal growing media. Then, 50 µM cordycepin and 1.5 nM pentostatin were added to spore suspensions separately, and in combination. This concentration of cordycepin was approximately equivalent to that of potato dextrose broth media of grown *C. militaris* control strain cultures (data not shown). Responsiveness (Figure 11), melanisation (Figure 12) and fungal emergence (Figure 13) from caterpillars were observed.

Responsiveness in control and degenerated strain-injected caterpillars was not significantly different, with or without cordycepin and pentostatin additions. All treatments involving spore injections had significantly lower responsiveness than their corresponding controls (df = 1, X^2^ = 142.5, 117.8, 142.5, 141.6, 110.9, 141.6, 86.19, 49.84, 84.85, 48.04, *p* values < 0.01).When both cordycepin and pentostatin were added to injections of degenerated strain spores, fungal emergence was significantly increased compared to the spores only treatment (df = 1, X^2^ = 11.94, *p* = 0.0468), although this was not the case for the addition of cordycepin or pentostatin alone (*p* values = 1.3026, 0.0936, respectively). The addition of cordycepin or pentostatin to control strain spores did not significantly change fungal emergence. These results suggest that the addition of cordycepin with pentostatin can enhance the pathogenic activity of a degenerated strain of *C. militaris*.

## 4. Discussion

In this study we demonstrate that the loss of cordycepin production observed during the repeated subcultivation of *C. militaris* occurs simultaneously with reductions in other metabolites and in the expression of genes related to sexual development. Reduction in pathogenic response in the model host *Galleria mellonella* caterpillars also takes place. Our work suggests that the pathogenic response to *C. militaris* by the host, developmental stage of the sexual phase of the fungus and its metabolite production are all affected by degeneration.

Successive subcultivation of the *CM2* isolate of *Cordyceps militaris* resulted in the development of the degenerated strain, which in LC-MS analyses showed significantly lower production of both cordycepin and pentostatin in potato dextrose agar-based cultures. This is consistent with the gene expression, with significantly lower levels of mRNA detected from the cordycepin-pentostatin biosynthesis cluster genes *Cns 1*, *3* and *4*. It is also consistent with the reduction of metabolite production from a metabolic pathway previously proposed to be associated with the production of cordycepin—purine metabolism. It is clear that during the cultivation of the fungus a major change in metabolite output has taken place.

In addition to these changes, there is evidence for a decline in the expression of genes responsible for sexual development—which suggests a shift from the characteristics of the teleomorph (sexual phase) towards the vegetative state which is facilitated by cultivation on agar. The velvet-related genes, which had lower levels of expression in the degenerated strain, are known to be conserved in ascomycetes [42]. *VeA* and *VosA* encode members of the velvet family of proteins, first discovered in the *Aspergillus* species, which play key roles in regulating sexual development and secondary metabolism in response to abiotic stimuli such as light [42]. The non-velvet protein master regulator gene known to control their expression is *LaeA*, a methyltransferase-encoding gene [43], which also had lower expression in the degenerated strain. Velvet proteins such as VeA, VelB and VelC (velvet proteins A, B, C) and VosA (variability of spores A) often act in complexes [43]. Therefore, changes have been shown in the high-level regulation of sexual development in the fungus, to have occurred simultaneously with the degeneration process. *SteA* and *Ste7* are genes which function downstream of the velvet proteins in sexual development, encoding a transcription factor and a MAP kinase, respectively. Overall, the gene expression data indicate that expression-based reprogramming of the fungus has occurred during degeneration. As we detect changes in a methyltransferase (*LaeA*), we suggest that epigenetic changes could play a role in this.

Significantly lower proportions of both full melanisation of, and fungal hyphal emergence from *Galleria mellonella* caterpillars injected with the degenerated strain suggest it displays a reduced pathogenic response in the host compared to the parental control strain. Both full melanisation, an observable insect immune response, and fungal growth, observed by hyphal emergence, differ between the strains. It is possible that this is driven by the lower production of cordycepin and pentostatin in the degenerate culture, but it is not unlikely that other factors also play a role, since there are multiple determinants of virulence in EPF [39]. 

There were limitations in the use of the *Galleria mellonella* model—such as the inability to assess point of death in the host. Observations of responsiveness, a measure for moribund status of caterpillars rather than a proxy for survival—did not show significant differences between the two strains. Melanisation, occurring later than loss of responsiveness in the majority of caterpillars, is an indication of caterpillars staying alive when unresponsive, as they were able to perform this immune reaction. Completion of full melanisation may be a better proxy measure for terminal infection by the fungus, with death most likely to have occurred between the onset and completion of melanisation. The findings from melanisation and fungal emergence data—measures of insect immune response, and fungal proliferation in the host, respectively—reveal significant effects of degeneration on pathogenic response to the *C. militaris* isolate.

The positive effects of added combined cordycepin and pentostatin on the success of washed degenerated strain spores in infecting caterpillars—shown by fungal emergence observations—indicate that these metabolites may confer a pathogenic advantage to the fungus over the insect host. This is of interest considering that previous studies have also shown the addition of cordycepin to confer an advantage to *C. militaris* and even other pathogenic species over the host [11,12]. In both *Galleria* injection experiments, in some caterpillars’ melanisation response failed to prevent fungal emergence, with observed outgrowth of hyphae through the cuticle after host death being a demonstration of successful infection [39]. For example, when injected with conidia from and spent PDB media from parental control strain cultures, almost all (97%) of the caterpillars showed full melanisation, and almost half (48%) of the total showed fungal emergence (Figure 9 and Figure 10). Unlike some instances of caterpillars seen before with melanised appearance [44], these caterpillars had a firm texture and did not exhibit symptoms of bacterial colonisation and decomposition of the host body. It is possible that secondary metabolites including cordycepin and pentostatin controlled infection in hosts by opportunistic bacteria in these melanised caterpillars, in the favour of the fungal pathogen and facilitating its colonisation, or that this was a result of the highly virulent nature of these fungal strains. 

Caterpillar gene expression assays following injection using curdlan as a mock fungal infection indicate that cordycepin indeed plays a role in repressing the insect immune system. Upregulated *Lysozyme* expression, induced by stimulation from curdlan, was reduced by addition of cordycepin to the injection serum. This effect provides evidence for the immune response-suppressing role of cordycepin in the context of an insect infection by *C. militaris*.

This study has produced combined evidence of the simultaneous decline of cordycepin, pentostatin and associated metabolites output, with sexual development gene expression of, and pathogenic response to a novel *Cordyceps militaris* isolate by degeneration following repeated subcultivation. Evidence also points to the biological purpose of cordycepin and pentostatin synthesis to aid host infection leading to the sexual development of *C. militaris*. Improving our understanding of the virulence mechanisms used by hypocrealean EPF underpins the development of this group of pathogens as biocontrol agents of economically important crop pests, as well as providing fundamental insights into host-pathogen coevolution.

## Figures and Tables

**Figure 1 microorganisms-09-01559-f001:**
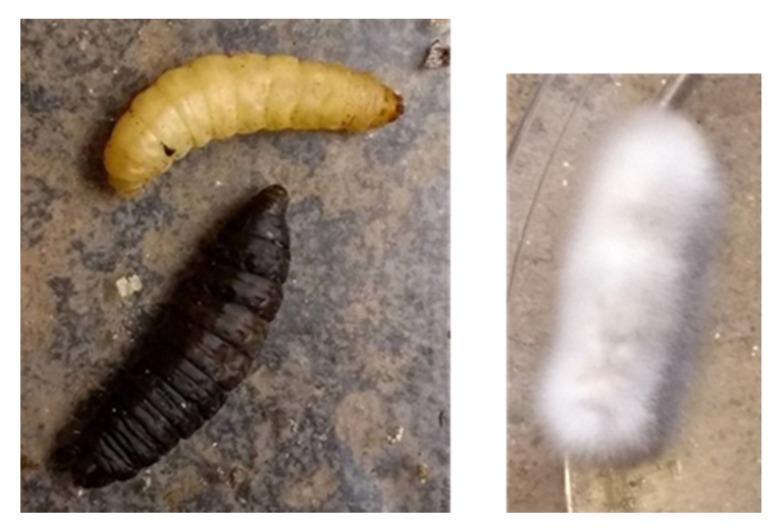
Melanised (**left**, **bottom**) and non-melanised (**left**, **top**) *Galleria mellonella* caterpillars, as well as one showing complete fungal emergence (**right**).

**Figure 2 microorganisms-09-01559-f002:**
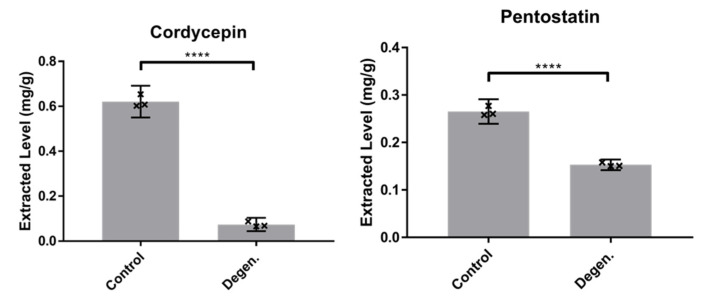
Cordycepin and pentostatin levels of control and degenerated *CM2* strains as detected by LC-MS analysis. Levels are presented in mg/g fresh fungal material extractions from PDA (potato dextrose agar) plates. Error bars show 95% confidence intervals, and **** labels show significant differences (*p* < 0.0001) as calculated using *t*-tests, with three biological replicates.

**Figure 3 microorganisms-09-01559-f003:**
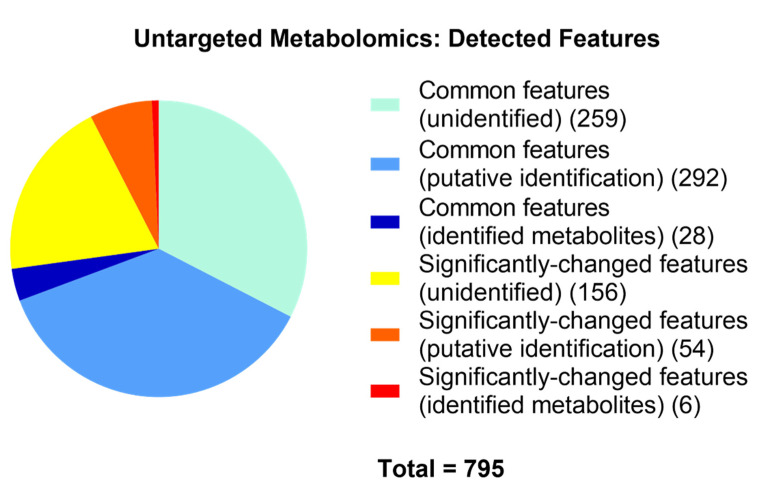
Detected features from base peaks, including identified and putatively-identified metabolites by untargeted metabolomics. Significant differences were determined by both t-tests with FDR corrections and having VIP values >1.

**Figure 4 microorganisms-09-01559-f004:**
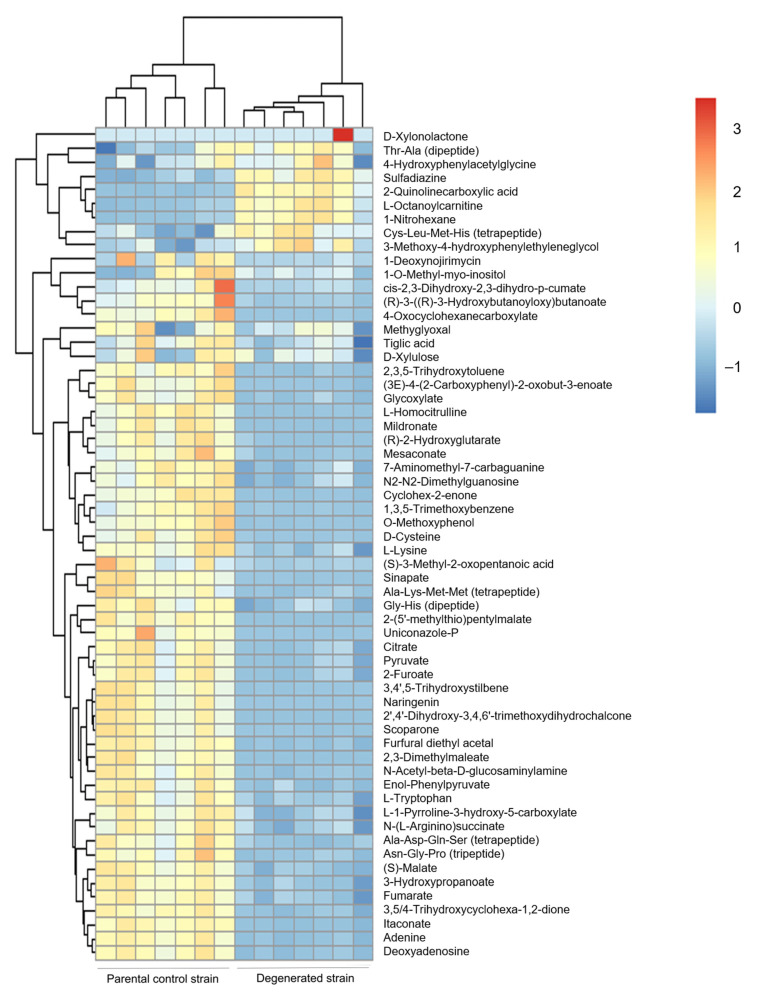
Heatmap representing untargeted LC-MS data from the parental control and degenerated strains for metabolites with significantly different levels. Significantly different metabolites were determined by both t-test with FDR correction and having VIP values >1. This figure was constructed using the heatmap package in R software.

**Figure 5 microorganisms-09-01559-f005:**
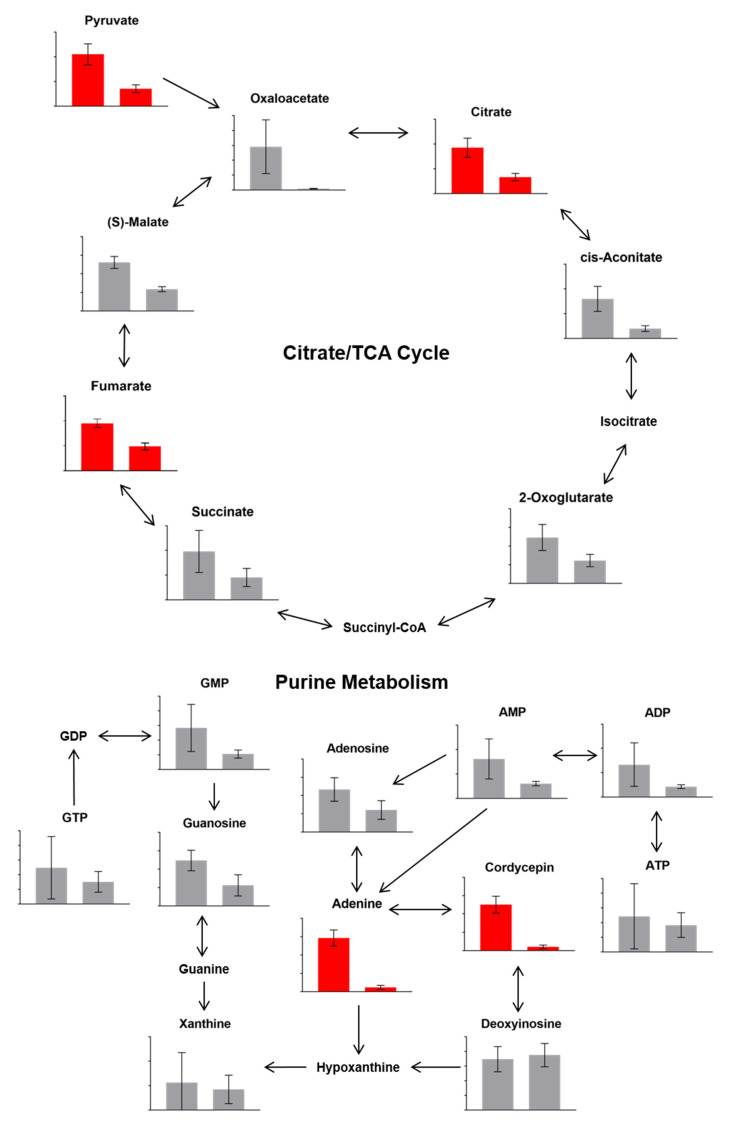
Metabolic pathways as described in the KEGG Database, with mean relative values of detection for parental control strain (left bar in graphs) and degenerated strain (right bar in graphs) samples. Graphs highlighted in red indicate significant differences between the parental control and degenerated strain samples, as determined by both t-test with FDR correction, and having VIP values > 1. Error bars show 95% confidence intervals. Arrows between metabolites present pathway steps.

**Figure 6 microorganisms-09-01559-f006:**
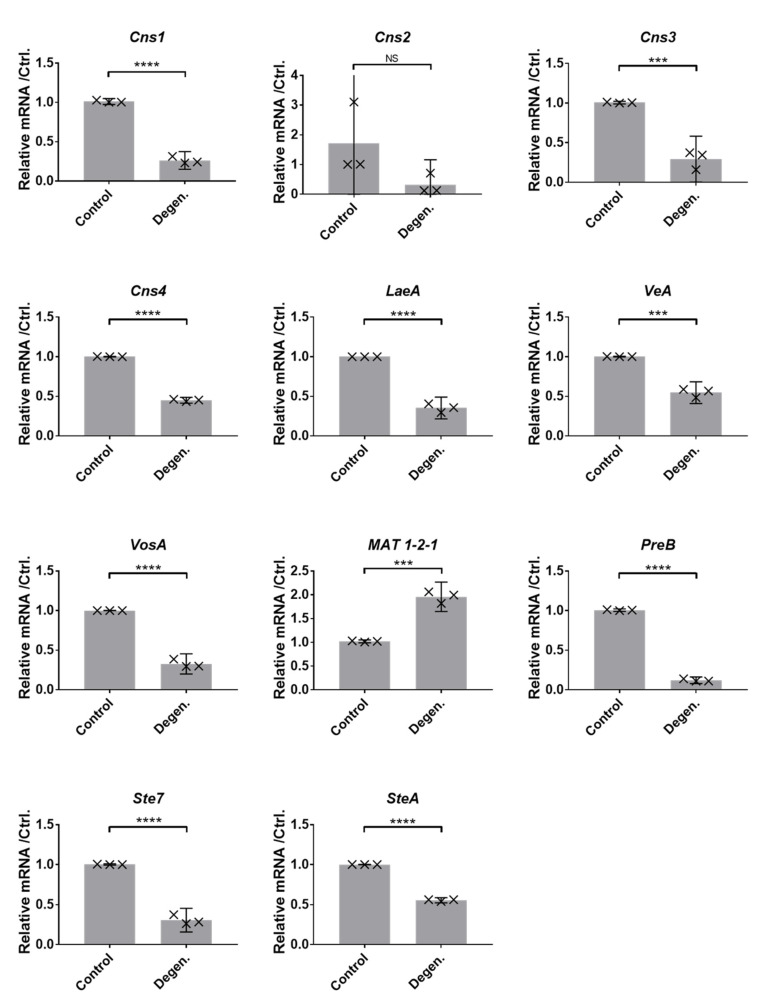
Relative mRNA levels of various genes of interest in parental control and degenerated *CM2* strains cultured on PDA (potato dextrose agar). Gene expression was normalised against the average of four controls, *GAPDH*, *Actin*, *Calmodulin* and *rps3*. Error bars show 95% confidence intervals, and labels show differences between means as calculated using *t*-tests. *** *p* < 0.001; **** *p* < 0.0001. Genes included are the cordycepin and pentostatin synthesis gene cluster (*Cns1-4*), velvet and related genes (*LaeA*, *VeA*, *VosA*), or other sexual development-related genes (*MAT 1-2-1*, *PreB*, *Ste7*, *SteA*). Three biological replicates, with each point representing an average of three technical replicates are shown.

**Figure 7 microorganisms-09-01559-f007:**
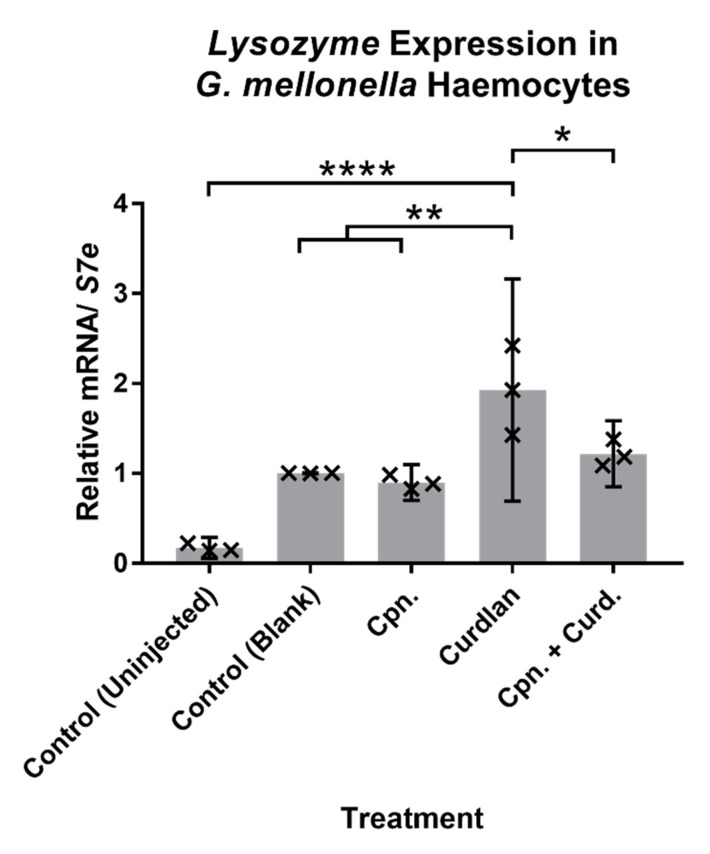
*Lysozyme* expression in *Galleria mellonella* haemocytes following injection by curdlan, with or without cordycepin treatment. Curdlan = curd; cordycepin = cpn. Expression levels are relative to the control gene *S7e*. Error bars show 95% confidence intervals, and labels show differences between means as calculated using t-test, with Bonferroni corrections. * *p* < 0.05; ** *p* < 0.01; **** *p* < 0.0001. Three biological replicates, with each point representing an average from three technical replicates are shown.

**Figure 8 microorganisms-09-01559-f008:**
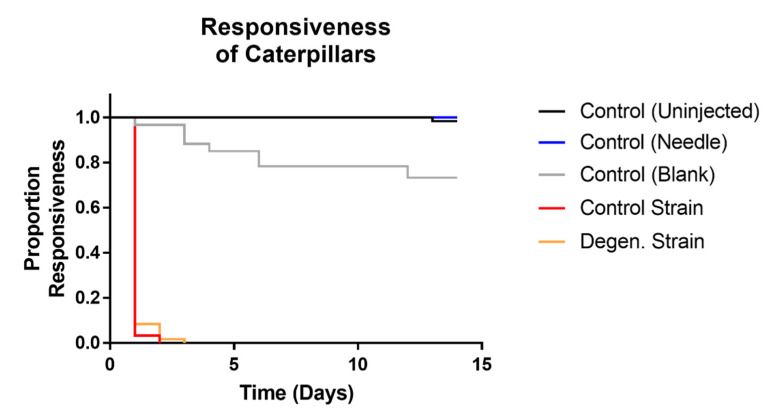
Kaplan Meier plot showing responsiveness observed in *Galleria mellonella* caterpillars following injections with *C. militaris* conidia and media from PBD cultures. Responsive caterpillars showed self-movement when disturbed individually using tweezers. Results were recorded on a daily basis over a period of two weeks, following injection with media and conidia from *CM2* potato dextrose broth cultures, as well as controls. Sixty caterpillars were used in each treatment.

**Figure 9 microorganisms-09-01559-f009:**
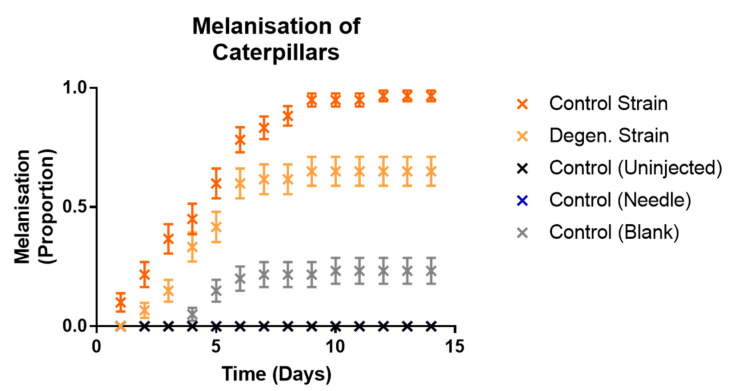
Proportions of fully melanised caterpillars observed in *Galleria mellonella* caterpillars following injections with *C. militaris* conidia and media from PDB (potato dextrose broth) cultures. Results were recorded on a daily basis over a period of two weeks, following injection with media and conidia from *CM2* potato dextrose broth cultures, as well as controls. Sixty caterpillars were used in each treatment. Error bars show 95% confidence intervals. Note the values for the control (needle) treatment are the same as control (uninjected)—consistently at zero.

**Figure 10 microorganisms-09-01559-f010:**
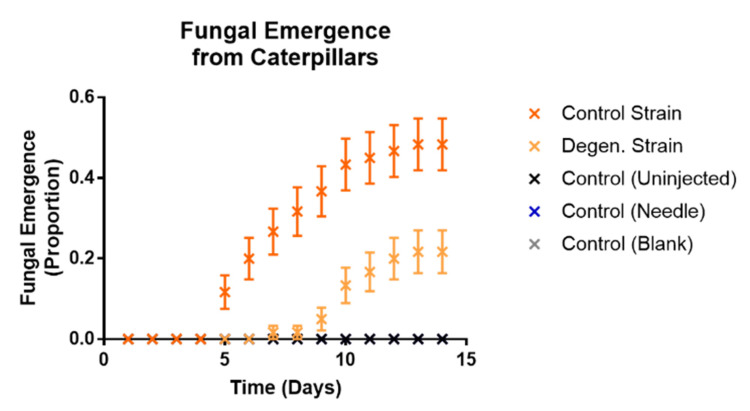
Proportions of fungal hyphal emergence observed from *Galleria mellonella* caterpillars following injections with *C. militaris* conidia and media from PDB (potato dextrose broth) cultures. Results were recorded on a daily basis over a period of two weeks, following injection with media and conidia from *CM2* potato dextrose broth cultures, as well as controls. Sixty caterpillars were used in each treatment. Error bars show 95% confidence intervals. Note the values for control (blank) and control (needle) treatments are the same as control (uninjected)—consistently at zero.

**Figure 11 microorganisms-09-01559-f011:**
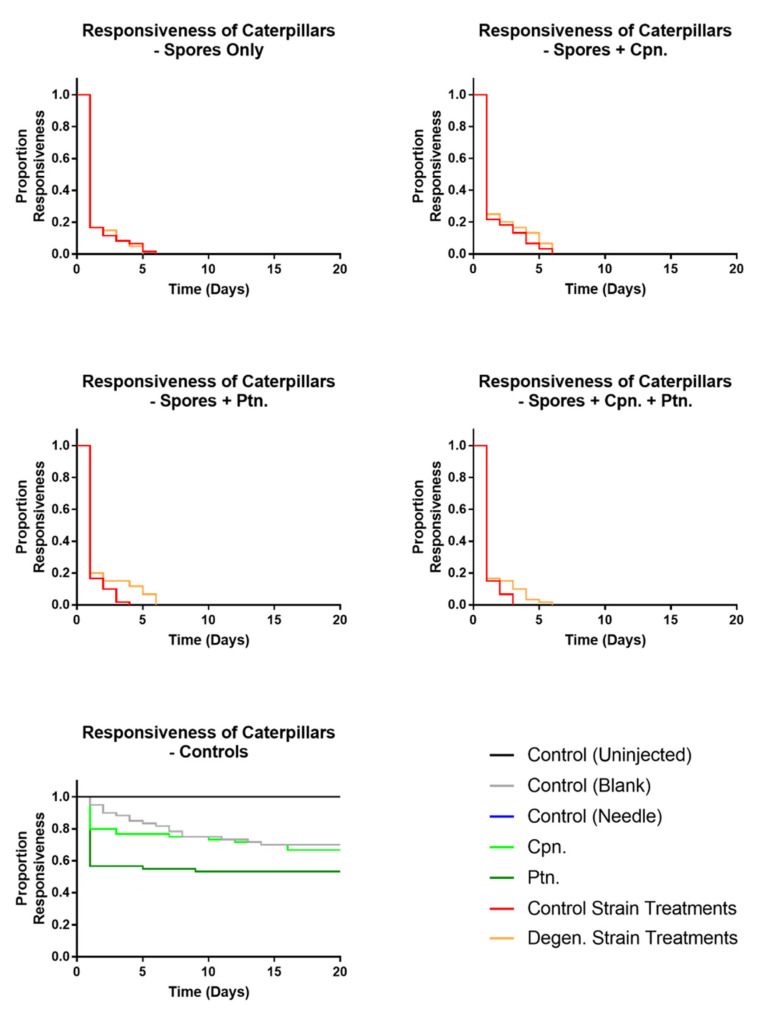
Kaplan Meier plots showing responsiveness observed in *Galleria mellonella* caterpillars following injections with washed *C. militaris* conidia, cordycepin and pentostatin. Responsive caterpillars showed self-movement when disturbed individually using tweezers. Results were recorded on a daily basis over a period of two weeks, following injection of PBS-washed conidia from *CM2* potato dextrose broth cultures, with and without cordycepin and pentostatin, as well as controls. Sixty caterpillars were used in each treatment.

**Figure 12 microorganisms-09-01559-f012:**
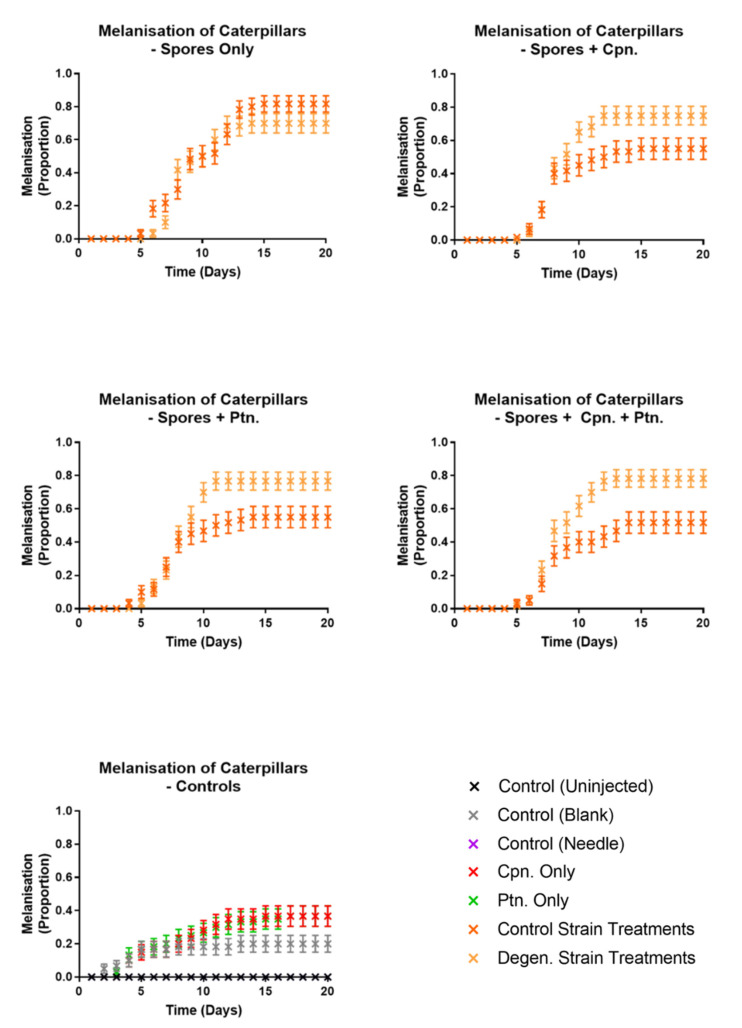
Proportions of fully melanised caterpillars observed in *Galleria mellonella* caterpillars following injections with washed *C. militaris* conidia, cordycepin and pentostatin—time course graphs. Results were recorded on a daily basis over a period of two weeks, following injection of PBS-washed conidia from *CM2* potato dextrose broth cultures, with and without cordycepin and pentostatin, as well as controls. Sixty caterpillars were used in each treatment. Error bars show 95% confidence intervals. Abbreviations: cpn—cordycepin; ptn—pentostatin. Note the values for the control (needle) treatment are the same as control (uninjected)—consistently at zero.

**Figure 13 microorganisms-09-01559-f013:**
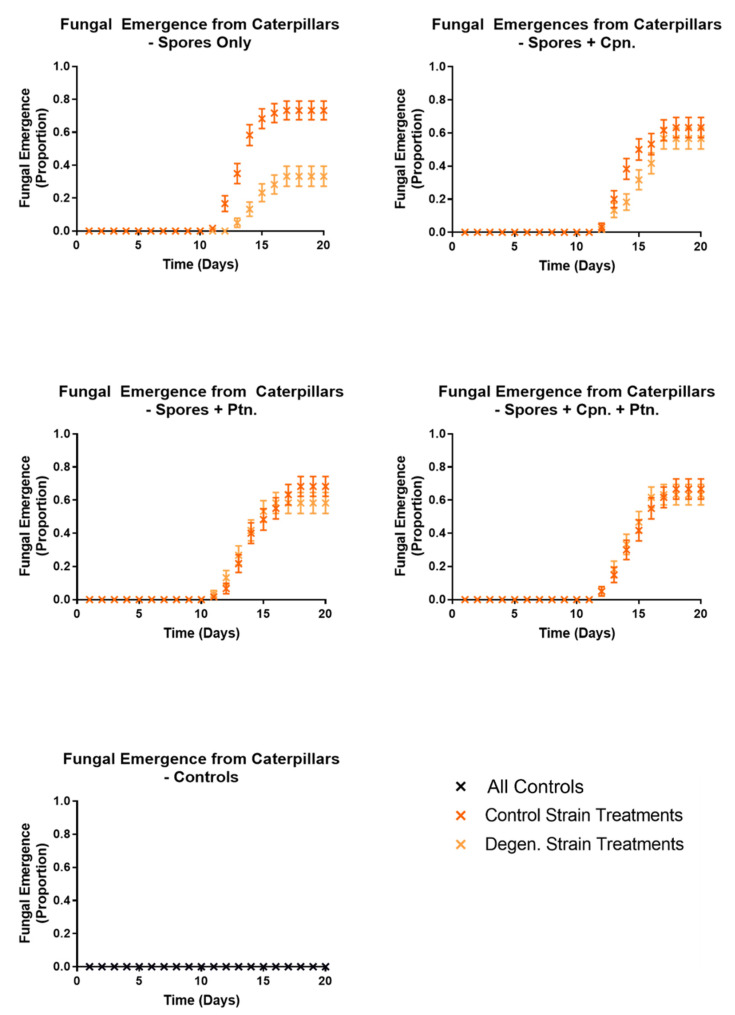
Proportions of fungal emergence observed in *Galleria mellonella* caterpillars following injections with washed *C. militaris* conidia, cordycepin and pentostatin—time course graphs. Results were recorded on a daily basis over a period of two weeks, following injection of PBS-washed conidia from *CM2* potato dextrose broth cultures, with and without cordycepin and pentostatin, as well as controls. Sixty caterpillars were used in each treatment. Error bars show 95% confidence intervals. Abbreviations: cpn—cordycepin; ptn—pentostatin. Note the controls were control (uninjected), control (blank), control (needle), cordycepin only and pentostatin only—all with zero values.

**Table 1 microorganisms-09-01559-t001:** Primers used in RT-qPCR analyses, to assess changes in gene expression from both, *Cordyceps militaris* and *Galleria mellonella*.

Gene	Sequences (5′-3′)	Species	Primer Reference
*Actin*	CGAAGACGTTGCCGCTCTGG CGGAGGCTGAGAATGCC	*C. militaris*	This study
*Calmodulin*	GCTTGCGCCCTCTCGCTGC CTCTGACTCGGAAGGGTTCTGGCC	*C. militaris*	This study
*Glyceraldehyde-3-Phosphate Dehydrogenase (GAPDH)*	CGTCAAGGTTGGCATCAACGG GCCGTTGACGACGAGATCG	*C. militaris*	This study
*Ribosomal Protein S3 (rps3)*	CCTTTCGCCCAAGAATAATTTAG GTTGAATGTAAAGATGTTTTTTGTAATCC	*C. militaris*	This study
*Cordycepin Synthesis 1 (Cns1)*	GCTTATCCGACTACATTTCCATCC CCAGCCGCTCCAGGTGC	*C. militaris*	This study
*Cordycepin Synthesis 2 (Cns2)*	CGCCGGTGTCCTCCAGAC CGAGGCGTGTGACACGC	*C. militaris*	This study
*Cordycepin Synthesis 3 (Cns3)*	CGAGTCAACCGCCTACACC GTAGGACTGGGGCAGCGG	*C. militaris*	This study
*Cordycepin Synthesis 4 (Cns4)*	GCCGGACAAAGAGAAACGAC CCAAGAGCATCTCTCCCGG	*C. militaris*	This study
*Loss of aflR-Expression A (LaeA)*	GGCTGTCGATCTGAACAAAATCC CGAGGGTAGCCAATGAATTTCGACC	*C. militaris*	This study
*Velvet A (VeA)*	CCAGTGCCCAGTGCCAGTTG GCAATGGGCGAGGGCGAG	*C. militaris*	This study
*Variability of Spores A (VosA)*	CTCTTCATACACATCACCAAAGGC CTGTTGGCGCACCTCAAGG	*C. militaris*	This study
*Mating type alpha pheromone receptor (PreB)*	CGTCGTTTGACCGCTTCGC CGACGCAGAGCGCGAG	*C. militaris*	This study
*MAT 1-2-1*	CTCAGTATCGACGGTCTCATCTACC CCAGTGCCGGACATCAAATGTCG	*C. militaris*	This study
*MAP kinase kinase Ste7 (Ste7)*	GATGGTCAACTCGAGATCGGG GTTGACAATGTAATCCGAGTGACAG	*C. militaris*	This study
*Transcription Factor SteA (SteA)*	GCTTTCTGCTCCCTACCGG GAAAGTCGAGAAACTGGCTCTTG	*C. militaris*	This study
*Ribosomal Gene S7e (S7e)*	TCCCAACTCTTGACCGACGA AGTGGTTGCGCCATCCATAC	*G. mellonella*	[22]
*Lysozyme (Lyso)*	ATGTGCCAATGCCCAAGTTG GTGGCTAGGCTTGGGAAGAAT	*G. mellonella*	[21]
*Gallerimycin (Gale)*	TATCATTGGCCTTCTTGGCTG GCACTCGTAAAATACACATCCGG	*G. mellonella*	[22]
*Galiomicin (Gali)*	TCGTATCGTCACCGCAAAATG GCCGCAATGACCACCTTTATA	*G. mellonella*	[23]
*Insect metalloproteinase inhibitor (IMPI)*	AGATGGCTATGCAAGGGATG AGGACCTGTGCAGCATTTCT	*G. mellonella*	[21]

For *C. militaris* gene expression analysis, cordycepin and pentostatin synthesis-related genes (*Cns1-4*) and sexual development-related genes (*LaeA*, *VeA*, *VosA*, *MAT 1-2-1*, *PreB*, *SteA* and *Ste7*) were analysed, whereby *Actin*, *Calmodulin*, *GAPDH* and *rps3* served as reference genes. For *G. mellonella* gene expression analysis, *Lyso*, *Gale*, *Gali* and *IMPI* were selected, and *S7e* was used as a reference gene. In all RT-qPCRs, three biological replicates were taken, with three technical replicates for each.

## Data Availability

Data is available at the Nottingham Research Data Repository (https://rdmc.nottingham.ac.uk/) and at MetaboLights (https://www.ebi.ac.uk/metabolights/).

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
