# Peer review of "Culture Degeneration Reduces Sex-Related Gene Expression, Alters Metabolite Production and Reduces Insect Pathogenic Response in Cordyceps militaris"

_microorganisms, 2021, doi:10.3390/microorganisms9081559_

Round 1

Reviewer 1 Report

These are my main comments on the MS (microorganisms-1271445) entitled:“ Culture degeneration reduces sex-related gene expression, alters metabolite production and reduces insect pathogenicity in Cordyceps militaris

It is a very interesting and well-written study investigating several pathogenicity effects of a well-known entomopathogenic fungi. Such studies are useful to enhance the use of biocontrol agents against insect pests.

My suggestion is to be published after minor revision. Authors may follow my suggestions to improve their MS before publication.

Comments

Line 148. Please provide the model of the LC-MS system.

Line 196. Please describe the method of the injection. What equipment was used?

Line 200. Replace “repeats” with “replications”

Line 208. Provide a reference for the software used for statistical analysis.

Line 233-235. When stating that differences are significant, values of df, F and P should be also presented. Please check this throughout the whole MS.

Figs. When there are more than one charts in a Fig they should be named with A, B etc.

Figs.7 & 8. Use A and B to mention the two graphs. In the 7A and 8A (upper) charts I cannot see the blue marks of Control (needle). Is this because they are identical to some other treatment?

Fig 9a, 10a. Please add the meaning of Cpn, Ptn etc in the legend. Figs should be self-explanatory without needing to search in the text for the meaning of various terms. There are many marks in the label that are missing from the charts, obviously because there was nothing to measure in these treatments. However, this can be a little confusing for the reader. Perhaps an explanation should be added in the legend.

Discussion is well written and includes many relevant previous publications.

Reviewer 2 Report

This manuscript describes the culture degeneration of entomopathogenic fungus Cordyceps militaris. This worldwide distributed fungus is a natural pathogen of lepidopteran and other insects. The authors describe the effect of repeated cultivation under laboratory conditions showing decreasing levels of cordycepin and other metabolites as well as the reduced expression of genes related to sexual development. The story is quite clear and results can be expected, authors used up to date methods of molecular biology and bioinformatics. But I have serious comments to the part related to experiments with alternative insect host – Galleria mellonella. This classical laboratory insect model develops in seven instars and the weight of 250-290mg (line 195) for sure doesn’t represent the fourth instar but probably the seventh. Or really the fourth instar with another weight? There is mentioned that 100µl was injected into the larvae (line 196) but this volume is unreal, normally 10-20µl is injected into the last instar larvae; 100µl is larger volume than whole amount of haemolymph in one larva, it means cause large stress reaction. I have also serious doubts about leaking out from the body during injection of such a large volume. Why was not possible to concentrate fungal samples (injected concentration is not shown in method section!), G. mellonella is very sensitive host to low doses of pathogens.... There is not mentioned the composition of artificial diet for G. mellonella larvae and also if they were fed during the experiment which took following 15 days (e.g. Fig. 7), they will die without food in five days and if last instars were used than they will pupate. Optimal rearing temperature for G. mellonella is around 30°C, but the experiment was at 22°C, so there can be the effect of the temperature as well. Another discussed part is about melanisation reaction, it is not only immune but also physiological reaction as result of activation of prophenoloxidase cascade. This cascade is not mentioned at all and also the method of melanisation measurement including three controls mentioned in Fig. 7. In graphs it is as proportion of melanisation - how it was quantified? It leads to 1.0 (Fig.  7) and thus fully melanised larvae survived another at least five days or were already dead?  Area under curve shows the same data as proportion of melanisation? If yes, only one type of graphs should be included into main text of the manuscript. Data presented in graphs shows often the same values for the triplicates, seems not very common situation.

Experiment with lysozyme expression in haemocytes of G. mellonella – how it was measured – also haemolymph without haemocytes contains lysozyme and according to methodology, haemocytes were not separate from the haemolymph.

In general, I don’t trust the part of effects on insect immunity.

I consider this manuscript not suitable for publication in Microorganisms.

Minor corrections:

  • Lines 204-205: “caterpillars were frozen in liquid nitrogen before RNA extraction from the haemolymph, two hours after injection” – The RNA was extracted from frozen haemolymph inside frozen whole larva? I can’t imagine how it can be done…
  • Figures use mix of 95% confidence interval and S.D.
  • Is there any rescue experiment showing that degenerated laboratory culture multiplied on insect host can get back its lost properties?

Round 2

Reviewer 2 Report

Authors improved significantly the manuscript according to my comments.

Some additional minor comments:

Line 18: “larval and pupal hosts (caterpillars)” – delete caterpillars or move towards to “larval”.

Line 194: “Injections of 100 μL“– still not mentioned what was injected, move this information from lines 357-359 to 194. “The injection suspensions were prepared in triplicate to avoid confounding effects from technical error in preparing them” - this is mentioned as response to reviewer but should be mentioned also in the manuscript.

Lines 357-359: “suspensions of conidia (100,000 spores) and liquid media from four-week old C. militaris CM2 strain potato-dextrose broth (PDB) cultures” – suspensions IN liquid media, how was this concentration selected, according to some dose-dependent experiment or reference? I still believe it is possible to concentrate it to lower volume than 100 µL for injection.

E.g. Fig. 9:  Proportions of melanisation – it is proportion (percentage) of fully melanised larvae, not proportion of melanisation when every larva can be partly melanised….
